# Feasibility of Casein to Record Stable Isotopic Variation of Cow Milk in New Zealand

**DOI:** 10.3390/molecules25163658

**Published:** 2020-08-11

**Authors:** Kavindra Wijenayake, Russell Frew, Kiri McComb, Robert Van Hale, Dianne Clarke

**Affiliations:** 1Department of Chemistry, University of Otago, P. O. Box 56, Dunedin 9054, New Zealand; kavindra.wijenayake@postgrad.otago.ac.nz (K.W.); kiri.mccomb@otago.ac.nz (K.M.); 2Isotrace NZ LTD, 167 High St, Dunedin 9016, New Zealand; robertv@chemistry.otago.ac.nz (R.V.H.); dclark@chemistry.otago.ac.nz (D.C.)

**Keywords:** stable isotopes, cow milk, *δ*^15^N signature, casein, climatic variables, IRMS

## Abstract

Dairy products occupy a special place among foods in contributing to a major part of our nutritional requirements, while also being prone to fraud. Hence, the verification of the authenticity of dairy products is of prime importance. Multiple stable isotopic studies have been undertaken that demonstrate the efficacy of this approach for the authentication of foodstuffs. However, the authentication of dairy products for geographic origin has been a challenge due to the complex interactions of geological and climatic drivers. This study applies stable isotope measurements of *δ*^2^H, *δ*^18^O, *δ*^13^C and *δ*^15^N values from casein to investigate the inherent geo-climatic variation across dairy farms from the South and North Islands of New Zealand. The stable isotopic ratios were measured for casein samples which had been separated from freeze-dried whole milk samples. As uniform feeding and fertilizer practices were applied throughout the sampling period, the subtropical (North Island) and temperate (South Island) climates were reflected in the variation of *δ*^13^C and *δ*^15^N. However, highly correlated *δ*^2^H and *δ*^18^O (r = 0.62, p = 6.64 × 10^−10^, α = 0.05) values did not differentiate climatic variation between Islands, but rather topographical locations. The highlight was the strong influence of δ^15^N towards explaining climatic variability, which could be important for further discussion.

## 1. Introduction

Milk is a significant part of the global diet and demand across the international food market is continually increasing [1]. The high consumption of dairy products has made this industry susceptible to fraudulent activity [2]. A significant part of this malpractice is misrepresenting the origin, a practice that is difficult to detect due to deficiencies in origin verification systems. As a result, determining the authenticity and origin of dairy products is vital for ensuring food safety in export markets and the global food trade. Stable isotopic methods have proven efficacy for detection of adulteration and mislabelling [3]. Several studies have discussed defining the origin of milk products using stable isotopic ratio analyses [4,5,6,7,8,9,10,11,12,13]. Of note is the study by Crittenden et al. [7], which demonstrated the possibility of identifying the geographical origin of milk in Australasia using multi-element stable isotope ratio analysis. 

Stable isotope ratios can be determined for many elements in nature; however, *δ*^2^H, *δ*^18^O, *δ*^13^C, *δ*^15^N, *δ*^34^S and ^87^Sr/^86^Sr are the isotopic measures that are commonly used in food forensic applications. These ratios vary across all four subsystems of the earth (hydrosphere, lithosphere, biosphere, atmosphere) through isotopic fractionation [12], due to environmentally based physiochemical and biochemical processes [14]. In terms of dairy products, previous studies have shown that a given isotopic ratio is mainly influenced by cattle feed as a function of the geological, climatic and agricultural surroundings within which they exist. Seasonality and climatic parameters such as rainfall, temperature, and humidity influence the values of *δ*^2^H and *δ*^18^O [3,7,8,12,15], while the *δ*^13^C values in milk, cheese and other products are mainly influenced by the photosynthetic origin of the feed type (C_3_, C_4_ and CAM—Crassulacean acid metabolism) [3,13,16,17,18], where the differences of CO_2_ fixation in photosynthetic processes [16] are reflected through *δ*^13^C values in milk [3,7,16,17]. Overall, plants (related to dairy farming feed) following C4 pathway have *δ*^13^C values from −10‰ to −20‰, while C3 plants vary from −22‰ to −34‰ [16,17,18,19,20,21]. The variation in *δ*^15^N is affected by several agricultural/soil conditions such as soil type, fertilizer practice, nitrate (NO_3_^-^) runoff (depending on the amount of regional rainfall), and the rate of nitrogen turnover [3,6,7,12,14,22]. Explanation of the fractionation in *δ*^34^S is more difficult when compared to other light stable isotopic systems, as *δ*^34^S is regulated by numerous physical processes (SO_2_ emissions, fertilizer practice, proximity towards coastal areas) in the region, rather than biological processes [7,10,14,15,23]. Strontium isotope ratios (^87^Sr/^86^Sr) have been considered to be independent of the effects of metabolic reactions, making it a robust tool for tracing the geological origin of dairy products [7,11,15,24,25].

Protein is a major constituent (ca. 30%) of the solid portion of milk [26], and casein is the main protein (60–90%) [27,28] and has been used as a target constituent for various milk studies. The main aim of this pilot study was to investigate the isotopic variation of *δ*^2^H, *δ*^18^O, *δ*^13^C and *δ*^15^N in casein extracted from bovine milk across New Zealand. As a country that experiences wide climatic fluctuations (from North Island—subtropical—to South Island—temperate and sub-Antarctic), New Zealand provides a natural laboratory for studies on environmental drivers. In previous studies of New Zealand bovine milk, Ehtesham et al. [8] showed the potential of *δ*^2^H and *δ*^13^C values measured from fatty acids and bulk milk powder for verifying the origin of milk production regions. In another study, by Ehtesham et al. [29], the influence of feed and water on the stable isotopic composition of milk was demonstrated. In recent studies outside of New Zealand, Huang et al. [30] elucidated the origin of milk products in the Chinese market, using the *δ*^2^H and *δ*^18^O values of milk water, while Garbaras et al. [3] showed the possibility of using the combined values of *δ*^18^O, *δ*^13^C and *δ*^15^N to authenticate cattle milk according to seasonal variation. The stable isotopic composition of casein has not been frequently used among studies for authenticating milk origin.

Milk production in New Zealand (NZ) has been an essential part of the national economy throughout the last century. Dairy exports from NZ were valued at NZ$ 16.6 billion, by the Dairy Companies Association of New Zealand, in 2018 [31]. Hence, establishing a regional database of stable isotopic composition for authenticating milk origin, is one of the benefits to secure the future of New Zealand milk export market from mislabeling and adulteration fraud.

## 2. Materials and Methods

### 2.1. Materials

In this study, milk samples were collected from 16 farms in New Zealand, ranging from North Island (8 farms) to South Island (8 farms), during the period from 26 March to 8 April 2018 (late summer-early autumn season). The sampling represents a snapshot in time, and no attempt was made to account for seasonal effects. Each farm consisted of mixed herds of cattle breeds, mainly Freshian and Jersy. The whole milk samples were collected from each farm within a short time frame of 1–2 days, and they were transported under chilled conditions to the laboratory in well-secured containers. Feeding practices in terms of the dairy industry in New Zealand are largely dominated by pasture-fed agriculture. Pasture growth in New Zealand is favoured due to its temperate climate and geography. Pasture represents almost all (95%) of the bovine feed in New Zealand dairy farms [32], C3 grasses predominate in all pastoral systems in New Zealand, as the growth of C4 grass is limited by the lower temperatures [33]. Crush and Rowarth [34] reported the presence of a C4 grass distribution in some parts of the North Island during mid-summer and into autumn, while no presence was observed in the South Island. Supplementary maize silage (C4 plant) is included as a part of the diet for cattle among some farms in North Island during summer but mainly used in the autumn-winter periods. The farm locations that were used in this study are given below (Table 1).

### 2.2. Preparation of Casein Samples

All milk samples were freeze-dried to constant mass. Freeze drying procedures were carried out in a Free-Zone freeze dryer (Labconco, Kansas City, MO, USA) at a temperature below −70 °C, and a vacuum pressure lower than 133 mbar. The freeze-dried milk powder samples were defatted using a modified Bligh and Dyer cold extraction procedure [8,35]. The aqueous layer from the final stage of the defatting method was taken for casein separation. Casein was separated by precipitation at its isoelectric point (pH = 4.6). Precipitation was achieved by the addition of 145 μL of 6 M (mol dm^−3^) hydrochloric acid (HCl) to the aqueous portion and vortexing for 1 minute. The mixture was allowed to stand for 10 minutes at room temperature before centrifugation was carried out at 3000 g for 30 min (at 20 °C) to separate the casein micelles. The remaining supernatant was discarded, and the separated casein samples were freeze-dried to remove any residual water. The samples were stored at −20 °C prior to isotope ratio analysis. 

### 2.3. Ethical Approval

Ethical approval was not needed for this research.

### 2.4. Bulk Analysis of δ^2^H, δ^18^O, δ^13^C and δ^15^N in Casein 

Each freeze-dried casein sample was homogenized to a powder before all measurement procedures. For *δ*^2^H and *δ*^18^O measurements, small aliquots of 0.6 ± 0.1 mg were weighed into 3.2 × 4 mm silver capsules followed by six days of equilibration at room temperature under laboratory atmospheric conditions [36]. The encapsulated samples were then vacuum dried at 50 °C for four days. Once dried the encapsulated samples were transferred immediately to the auto-sampler (Costech^®^ zero blank auto-sampler-Costech Analytical Technologies, Firenze, Italy), connected to a Temperature Conversion/Elemental Analyser (TC/EA) coupled with a Delta V Advantage (via ConFlo-III Interface) Isotope Ratio Mass Spectrometer—IRMS (Thermo-Finnigan, Bremen, Germany). The samples were converted to H_2_ and CO in the TC/EA reduction furnace, via thermolysis over glassy carbon at 1400 °C. The respective gases were separated through a packed Gas Chromatography (GC) column before passing to the IRMS. The final measurements obtained from the IRMS were calibrated and normalized to the VSMOW-SLAP scale (Vienna Standard Mean Ocean Water–Standard Light Antarctic Precipitation) using USGS 53 (*δ*^2^H_VSMOW_ = +40.2‰, *δ*^18^O_VSMOW_ = +5.47‰; U.S Geological Survey, Reston, VA, USA), and USGS 47 (*δ*^2^H_VSMOW_ = −150.2‰, *δ*^18^O_VSMOW_ = −19.80‰; U.S Geological Survey, Reston, VA, USA) which were measured concurrently with the samples. Instrumental drift correction was performed using duplicates of an in-house casein quality control material (*δ*^2^H_VSMOW_ = −112.1‰, *δ*^18^O_VSMOW_ = +12.1‰) measured in duplicate between every 12 samples.

For *δ*^13^C and *δ*^15^N measurements, the sample preparations before weighing were similar to those stated above. Aliquots of 0.8 ± 0.1 mg were weighed into 3.5 × 5 mm tin capsules and stored in desiccators until measurement was carried out as described previously [29]. During the analysis, the samples were combusted into CO_2_ and N_2_, in an elemental analyzer. The isotopic ratios of the respective gases were measured under a continuous flow system using a Delta Advantage IRMS (Thermo-Finnigan, Bremen, Germany). Final raw data was normalized against international scales (Carbon–VPDB and Nitrogen–AIR) using two certified reference materials of Glutamic acid (USGS 40; *δ*^13^C = −26.39‰, *δ*^15^N = −4.52‰ and USGS 41; *δ*^13^C = +37.63‰, *δ*^15^N = +47.57‰) and an EDTA laboratory quality control material (Elemental Microanalysis Ltd., Cornwall, UK). Isotopic values of carbon and nitrogen for EDTA are −38.52‰ and −0.73‰, respectively. Time-based instrumental drift was corrected for using the EDTA isotopic values measured between every 12 samples. Linearity correction was performed in a similar manner using EDTA standards. Furthermore, EDTA measurements (*n* = 16) were used for the determination of analytical precision, which was ±0.20‰ for *δ*^15^N and ±0.23‰ for *δ*^13^C. 

All isotopic measurements of casein samples were used to determine the power of geographical discrimination based on the origins of farms in South Island and North Island. The overall casein data spread is presented using Box-Whisker plots under each variable (*δ*^13^C, *δ*^15^N, *δ*^2^H and *δ*^18^O) in Figure 1. Figure 2 presents the isotopic values of all four variables plotted separately in correspondence to the geographic locations of all 16 samples/farms. Principal Component Analysis (PCA) was carried out as an unsupervised classification process, using the R statistical software, followed by a score-location map which was plotted using the sample scores derived from PCA analysis for each farm milk, under principal component 1 (PC-1) and principal component 2 (PC-2). Our data was scaled to z-scores prior to the Principal Component Analysis.

## 3. Results and Discussion

Important differences in *δ*^13^C and *δ*^15^N values were observed using Box-Whisker charts (Figure 1). The charts show the variability of casein isotope measurements (*δ*^2^H, *δ*^15^N, *δ*^18^O and *δ*^13^C) between North Island (NI) and South Island (SI) farms. The *δ*^13^C values for the SI farms were similar, the median value of −27.5‰ and range from −28.4‰ to −26.9‰. The NI farms, which had a median value of −24.3‰ and a broader range from −26.8‰ to −16.9‰. Apart from one outlying farm, the majority of the *δ*^13^C measurements exhibit values characteristic of a diet of C3 plants, the outlier NI farm exhibits a *δ*^13^C value of −16.9‰ which suggest the use of a C4 plant-based supplementary feed. The highest *δ*^15^N values were recorded from the NI farms (5.0‰ to 7.6‰) compared to lower values for the SI farms (3.8‰ to 6.3‰). As agricultural practices on the chosen farms were similar, the difference in *δ*^15^N indicates an intrinsic difference related to the ^15^N/^14^N fractionation process which could be exploited as an origin indicator. Based on these results, the isotopic values of *δ*^13^C and *δ*^15^N are capable of significantly differentiating dairy from the two islands (Mann-Whitney U test, *p* < 0.05).

Past studies [3,7,8,15,37], have shown that *δ*^18^O and *δ*^2^H values in milk powders mainly depend on the climatic conditions, defining the surroundings of the farming locations. However, *δ*^2^H is prone to indicating larger variations than *δ*^18^O in correspondence to climatic changes creating a decoupling between the two. *δ*^2^H values respond to isotopic fractionations regarding the water that is consumed by the animals and plants combined with photosynthetic effects; whereas *δ*^18^O values are also affected by the animal, and plant metabolism processes as a portion is lost as CO_2(g)_. Consequently, the difference in the medians between the two Islands for *δ*^18^O (0.4‰) and *δ*^2^H (4.1‰) was evaluated to be insignificant (Mann-Whitney U test for *δ*^2^H, *p* = 0.38, α > 0.05; and for *δ*^18^O, *p* = 0.34, α > 0.05). Therefore, individual values of *δ*^18^O and *δ*^2^H could not differentiate milk from the two islands. 

Overall, these stable isotopic distributions can be explicitly viewed with correspondence to each variable and geographical location (Figure 2). The colour scales were created separately for each isotopic component (*δ*^13^C, *δ*^15^N, *δ*^2^H and *δ*^18^O) in order to illustrate each distinctive isotopic spread. Accordingly, a strong variance can be seen in the *δ*^15^N isoscape in correlation to latitudinal gradient than any of the investigated isotopic systems. Soil *δ*^15^N variation could be a major contributor to the observed casein *δ*^15^N values. However, our measurements of soil *δ*^15^N, presented in the Appendix A show a very weak relationship with latitude (*r^2^* = −0.05, Appendix A) and with casein *δ*^15^N (*r^2^* = 0.12, Appendix A). Hence the soil *δ*^15^N is a minor driver of casein *δ*^15^N in this case. The clear trend between South Island and North Island samples displayed in the absence of a similar trend in soil *δ*^15^N shows that casein *δ*^15^N could be investigated further as a geo-climatic tool.

As a result of the profound changes in climatic parameters such as rainfall and wind that North Island and South Island experience seasonally, defining *δ*^2^H and *δ*^18^O values using casein can become complicated. Additionally, analyzing the climatic effects in terms of casein *δ*^2^H and *δ*^18^O can become difficult due to the higher degree of metabolic effects. 

Casein samples from the respective farms (*n* = 16) could be separated as NI and SI in two groups based on a principal component analysis (PCA). This method of practice is in agreement with previous studies [3,7,8,10,14,38,39], by proving the efficacy of stable isotopic variables in distinguishing such geographical regions. 

This exploratory data analysis was important for understanding the distinguishing power of each variable when used in a collective arrangement. In the PCA bi-plot (Figure 3), principal component 1 (62%—original variance explained) and principal component 2 (21%—original variance explained) together explained an accumulated variance of ~83% in the overall data [40]. In this multivariate ordination technique, *δ*^13^C values) and *δ*^2^H values mainly contributed to PC-1 (PC-1 loadings of +0.545 and −0.503 respectively); *δ*^15^N values and *δ*^18^O values mainly contributed to PC-2 (PC-2 loadings of −0.681 and −0.486, respectively). Nevertheless, all four variables exhibit an absolute PCA loading close to 0.5 on PC-1 (Table 2). Whereas, *δ*^15^N was the only variable which showed an absolute loading higher than 0.5 on PC-2. From the biplot Figure 3), it appears that PC-2 is the main differentiator between the North and South Islands. Furthermore, as per the correlation matrix between variables; *δ*^13^C and *δ*^15^N show a strong correlation (*r^2^* = 0.64), followed by a similar correlation (*r^2^* = 0.62) between *δ*^18^O and *δ*^2^H. In contrast, *δ*^18^O and *δ*^2^H both showed moderate but inverse correlations with *δ*^13^C and *δ*^15^N. In correspondence with the above results, previous studies [16,41,42,43,44] have reported strong links between the above-stated pairs of variables. These results from PCA demonstrates the limitations of using casein as source material for investigations to determine the source origin of dairy [45,46].

A PCA score-location map (Figure 4) was determined for making inferences on the subjective relationship between the stable isotope values and geographical location. 

Due to *δ*^15^N being the prime component of PC-2, it explains most of the variance across the latitudinal gradient (vertical axis) of the location map. Accordingly, individual PC2 scores ranged between −1.24 to +1.93, indicating two farms from the North Island to be predicted as South Island farms. This uncertainty could be due to influences of loadings on PC-2 by the weakly correlated variables *δ*^18^O and *δ*^2^H. Moreover, stable isotopic distribution of *δ*^15^N that was shown using Figure 2d plot, shows more substantial similarities than shown in Figure 4. Therefore, as the South Island (9 °C to 11 °C) and North Island (16 °C to 18 °C) had different daily temperatures during the sampling periods, this could be an indication that temperature may influence casein *δ*^15^N values. Furthermore, individual PC-1 scores can be related to the longitudinal gradient of the map in order to explain the horizontal variance of the respective scores (−2.59 to 3.90). Even though *δ*^2^H and *δ*^13^C are the main drivers (highest PC-1 loadings in Table 2), the farms are not significantly separated in comparison to PC-2. This lack of separation could be due to the effect of multiple correlations among all four variables as well as the variation of eco-geographic variables [47] across all New Zealand farms, which can complicate the fractionation of stable isotopic ratios. Thus, according to this pilot study, *δ*^15^N and *δ*^13^C can be considered to be the major variables distinguishing South Island farms from North Island farms. However, in contrast to the North Island, the low variance in scores (PC-1 = 1.0; PC-2 = 0.4) for South Island farms explains the lower degree of applicability of this specific model to landscapes such as the South Island, where it is highly susceptible to climatic fluctuations [48,49]. 

Previous studies [7,16,21,50] have shown the utility of using *δ*^13^C casein as an alternative to skim milk *δ*^13^C towards determining the feed practice in cases of milk authenticity. Accordingly, *δ*^13^C values for casein samples were predicted to be in the lower range due to the dominance of C3 plants for summer feed. Casein *δ*^13^C for South Island and North Island farms ranged from −28.38‰ to −26.89‰ and −26.83‰ to −22.46‰, respectively. North Island casein *δ*^13^C values were comparable with the results of Crittenden et al. [7]. The outlier value (−16.95‰) in one of the North Island farms was found to be provided with a substantial proportion of maize silage mixed diet (C3 grass as a minority), which could have made a potential impact with enrichment on resulting *δ*^13^C values rather than depletion. Camin et al. [21] quantitatively discussed this phenomenon by reporting a positive shift of casein *δ*^13^C of 0.7–1‰ for every 10% addition of maize to the diet. However, dairy cattle in the remaining North Island farms were found to be pasture (C3 grass) fed, with a median value (−24.9‰) for casein A *δ*^13^C value below −23.5‰ is the threshold for C3 grass feed with an absence of maize [21,51]. The casein *δ*^13^C values here are enriched by ca. 1–4‰ (from diet to milk) during the metabolic transformation from diet to milk [51]. 

Regardless of the consistency among feed, *δ*^13^C values in milk can also be affected by the climatic conditions where the pasture was grown [52]. This could be the reason for the separation which was observed under *δ*^13^C measurements between South Island and North Island farms; as evapotranspiration and photosynthetic effects could encourage *δ*^13^C shifts under different temperature and humidity conditions [8,52]. In response, the narrow distribution of *δ*^13^C measurements for South Island farms could define its inherent temperate nature as well as milk regions dominated by grassland domination [50], while the wider distribution is an indication of the subtropical climates which North Island encounters in most parts of the year. Therefore, this demonstrates the value of using *δ*^13^C to differentiate farming regions in situations where the given feed type is consistent.

In comparison to *δ*^13^C, the isotopic fractionation of nitrogen is more complicated (nitrogen cycle-specific). The intensive use of fertilizers adds complexity by altering soil conditions which can be detected through the nitrogen uptake of plants [53]. Nevertheless, the differences in *δ*^15^N between C3 and C4 plants are imperceptible within the detection limits [19,54]. Hence, synthetic and natural fertilizers practices can easily be differentiated using *δ*^15^N. Bontempo et al. [51] and Crittenden et al. [7] reported on understanding the agricultural practice (pasture-raised cattle) through *δ*^15^N_casein_ of bovine milk, while Liu et al. [38] observed the link between fertilizer practice and *δ*^15^N_casein_ signatures of goat milk. However, several other studies have viewed this behaviour between feed type and milk *δ*^15^N in a more explicit manner. Among them, Knobbe et al. [19] proposed the importance of using urine *δ*^15^N rather than milk *δ*^15^N, towards understanding the feeding system.

Milk *δ*^15^N ratios were indistinguishable between the grass and maize diets, as reported by Camin et al. [21]. Camin also suggested that the resulting diet-animal product shift carries a lesser input from the specific dietary system but is more variable due to metabolic effects. Thus, the nutritional quality of the dietary ingredients (e.g., C/N ratio) could profoundly affect the isotopic ratios of nitrogen in animal tissues [55,56,57] as well as animal products [21]. 

Therefore, the intrinsic stability within New Zealand farming practices and agricultural environments (e.g., fertilizer practice) enables exploration of the possible relationships between *δ*^15^N and climatic conditions. As the South Island and North Island experience two distinctive temperature regimes, the majority of the contribution towards strong correlations between *δ*^15^N with *δ*^13^C and *δ*^15^N with PC-2 axis could be considered to be caused by climate (humidity and temperature) dependent variances. Additionally, isotopic fractionation of nitrogen related to dairy cows was estimated to be high in metabolic origin, but low in digestive origin [58]; which again outlines the wider possibility of *δ*^15^N being affected by temperature as metabolic rates of animals are indirectly affected by environmental temperatures (e.g., heat stress). However, further work would be necessary to prove this relationship in order to establish a more robust concept. The variance in casein *δ*^15^N within each island could be due to differences in soil conditions, amount of rainfall, soil age and other complex parameters [7]. 

A comprehensive list of sources has shown the correspondence between milk casein and milk water [21,38,51], as well as the closeness between milk water and drinking water, for *δ*^18^O and *δ*^2^H values [3,13,29,50]. In addition, other observed variations of *δ*^18^O and *δ*^2^H concerning milk authentication work, can be integrated when concerning the consequences of farming environments (e.g., climate effects and topography [7,13,38,59], vegetal/forage water [51,60], and feed type [29,51,61]. The cows in the present study were pasture grazed. The effect of vegetation or feed type effects on *δ*^18^O and *δ*^2^H of milk or casein is negligible [21,51]. The inverse correlation of *δ*^18^O and *δ*^2^H with *δ*^15^N with *δ*^13^C indicated that the isotopic fractionation of casein *δ*^18^O and *δ*^2^H could be strongly affected by geographical parameters such as altitude and distance from the evaporation source rather than climatic components (temperature, humidity) [8,38,51]. Since most of the sampled farms in the South Island were close to the South-East coast (near to evaporation source), their values were enriched in ^18^O and ^2^H (as reflected by the median values) compared to the North Island farms. The stable isotopic ratios of oxygen and hydrogen in casein were enriched in comparison to results from Ehtesham et al. [61] and Crittenden et al. [7] on New Zealand bulk milk. However, the casein *δ*^2^H distribution between Islands were not in agreement with the study by Ehtesham et al. The difference may be due to New Zealand’s inconsistent climate variability between seasons, which could cause short term fluctuations on casein *δ*^2^H values [7]. Therefore, in conditions where climatic and geographical variability is experienced, casein *δ*^18^O and *δ*^2^H would be weaker in differentiating farming locations.

Moreover, even though seasonal changes could affect the type of pasture growth (C3 and C4) which then could be observed through the variations in milk/casein *δ*^13^C, *δ*^15^N and *δ*^18^O [3], its effect would be considered insignificant in this study as the sampling process was carried out within less than two weeks.

## 4. Conclusions

In summary, this study demonstrated the potential held by stable isotope data from casein towards differentiating farming regions with different climatic environments (Sub-Antarctic and Sub-tropical climates). This approach was possible due to the predictable patterns that were observed through *δ*^15^N and *δ*^13^C, mainly based on temperature variations in correspondence with the latitudinal gradient. Steady maintenance of fertilizer practice and nutritional content (C/N ratio) in feed would allow *δ*^15^N fractionation in milk production to become highly influenced by climatic parameters such as temperature and humidity. Nevertheless, further work should be carried out to explore further this encouraging interaction between metabolic fractionation of *δ*^15^N and temperature. Such work could direct us towards exploring the possibilities in using *δ*^15^N in order to understand heat stress effects. 

Climatic variables did not as significantly influence casein *δ*^2^H and *δ*^18^O as much as geographical location. Farms near coastal areas appeared to be enriched in ^18^O and ^2^H resulting in higher isotopic ratios for casein. In terms of future work, the seasonality and topographical effect on casein *δ*^18^O and *δ*^2^H would be further investigated. Nevertheless, when solving provenance cases where it is profoundly affected by climatic variability, past work on the correlation of bovine milk *δ*^2^H with grass (feed) fatty acids and drinking water could be an effective approach [8,29,61], which we would recommend. The use of casein *δ*^15^N, *δ*^13^C, *δ*^18^O and *δ*^2^H to distinguish milk products over regions where feed type, farming environment and agriculture practice is similar would be inadequate. That may be achieved with the inclusion of ^87/86^Sr ratios in tandem with trace elements by creating a better model. 

## Figures and Tables

**Figure 1 molecules-25-03658-f001:**
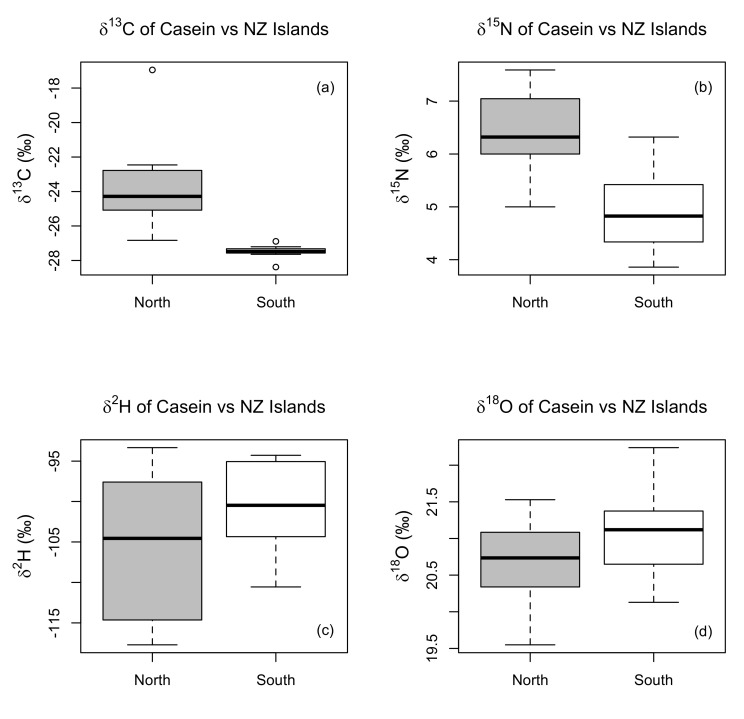
Stable isotopic distribution of casein samples across dairy farms in the North Island and the South Island of New Zealand: (**a**) *δ*^13^C values of casein vs NZ Islands; (**b**) *δ*^15^N values of casein vs NZ Islands; (**c**) *δ*^2^H values of casein vs NZ Islands; (**d**) *δ*^18^O values of casein vs NZ Islands.

**Figure 2 molecules-25-03658-f002:**
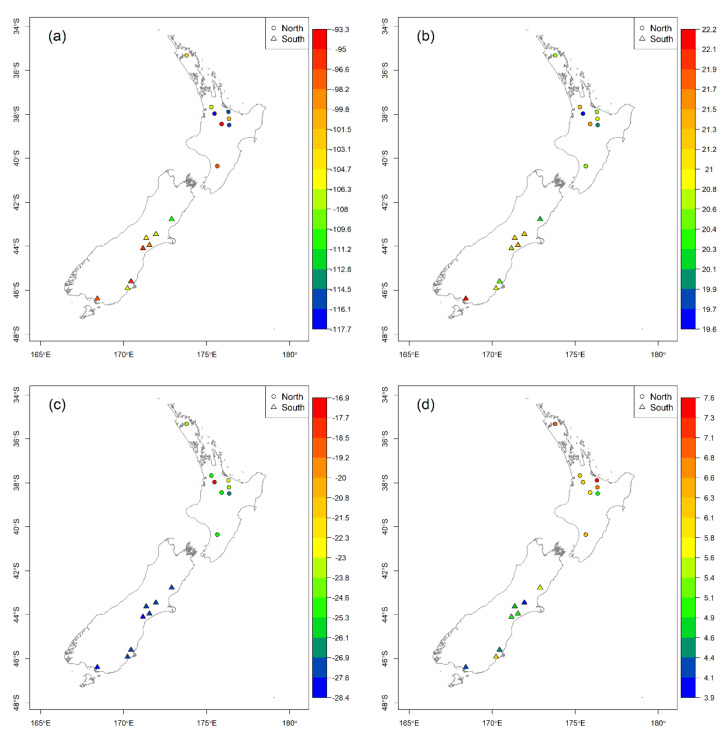
Distribution of stable isotopic values measured in casein samples from dairy farms across the North Island and South Island of New Zealand: (**a**) *δ*^2^H values of casein; (**b**) *δ*^18^O values of casein; (**c**) *δ*^13^C values of casein; (**d**) *δ*^15^N values of casein.

**Figure 3 molecules-25-03658-f003:**
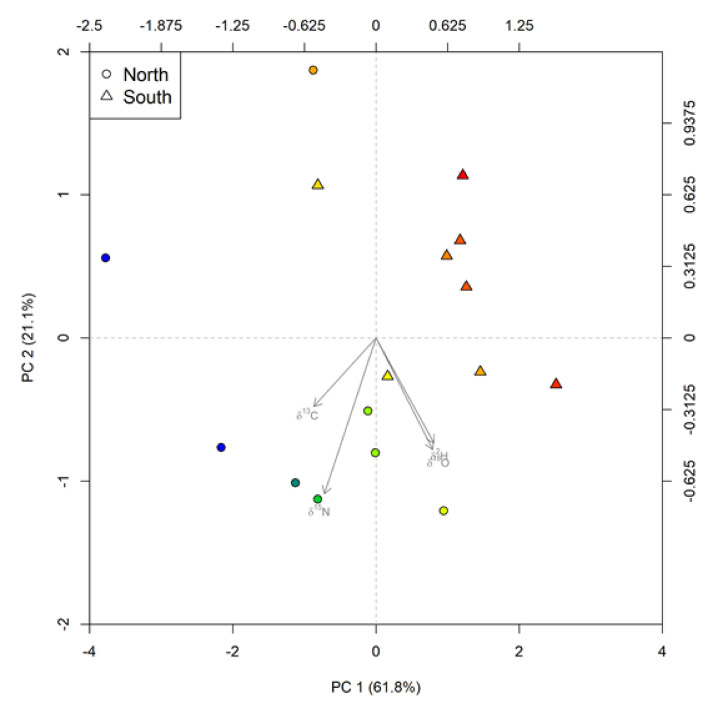
Principal Component Analysis through a Bi-plot presentation on New Zealand casein samples using four variables (*δ*^13^C, *δ*^15^N, *δ*^2^H and *δ*^18^O). The colour of individual points was determined from the average of PC1 and PC2 scores. As the scores of PC1 and PC2 increase from lowest to the highest, the colour frequency of the points increase from red to blue (in correspondence to a visible color spectrum).

**Figure 4 molecules-25-03658-f004:**
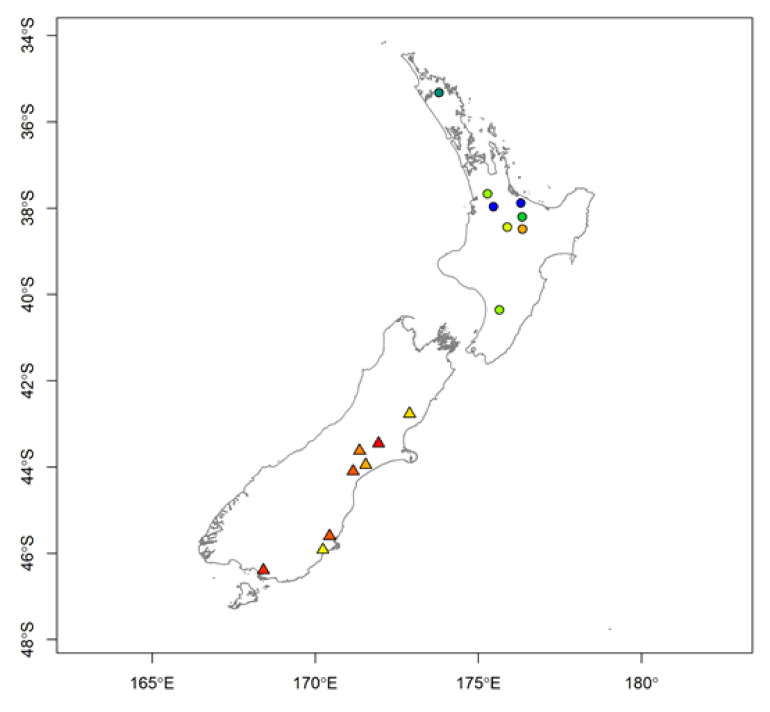
PCA score-location map of South Island and North Island dairy farms (16 sampling locations; using four variables—*δ*^13^C, *δ*^15^N, *δ*^2^H and *δ*^18^O) which uses the same system of colours that is used in Figure 3 to illustrate the spread in the average scores of PC1 and PC2 with the location of the farms.

**Table 1 molecules-25-03658-t001:** Geographical coordinates of the farming locations and the isotopic values of the casein samples (*δ*^2^H, *δ*^18^O, *δ*^15^N and *δ*^13^C).

Region	Latitude	Longitude	*δ*^2^H, ‰	*δ*^18^O, ‰	*δ*^15^N, ‰	*δ*^13^C, ‰
South	−45.9169	170.23	−103.90	21.03	6.32	−27.46
South	−44.0974	171.16	−94.78	20.75	4.95	−28.38
South	−43.4520	171.94	−104.77	21.22	3.86	−27.20
South	−42.7667	172.89	−110.55	20.13	5.76	−26.89
South	−46.3900	168.41	−95.34	22.24	4.20	−27.64
South	−45.5983	170.44	−94.28	20.55	4.47	−27.50
South	−43.6232	171.36	−103.45	21.21	4.70	−27.44
South	−43.9495	171.55	−97.48	21.53	5.08	−27.49
North	−37.8833	176.3	−114.49	20.77	7.59	−22.46
North	−38.4831	176.35	−114.79	20.00	5.00	−26.83
North	−37.9630	175.46	−117.70	19.55	6.29	−16.95
North	−35.3226	173.79	−102.87	20.68	7.20	−23.58
North	−38.4359	175.89	−93.34	21.53	5.87	−25.11
North	−40.3549	175.64	−95.10	20.70	6.35	−24.99
North	−37.6694	175.28	−106.23	21.36	6.13	−25.04
North	−38.2012	176.34	−100.08	20.81	6.89	−23.10

**Table 2 molecules-25-03658-t002:** PCA loadings for PC1 and PC2 in correspondence to the four variables (*δ*^2^H, *δ*^18^O, *δ*^15^N and *δ*^13^C) which were used in this study.

Variable	Principal Component 1	Principal Component 2
*δ*^2^H	−0.503	−0.459
*δ*^18^O	−0.495	−0.486
*δ*^15^N	+0.453	−0.681
*δ*^13^C	+0.545	−0.299

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
