# Peer review of "Feasibility of Casein to Record Stable Isotopic Variation of Cow Milk in New Zealand"

_molecules, 2020, doi:10.3390/molecules25163658_

Round 1

Reviewer 1 Report

The authors investigate 16 Casein samples extracted from milk from 16 different dairy farms in New Zealand during late summer early autumn. They investigate the stable isotope ratios and interpret them with respect to geographic origin.

I see several shortcomings in the manuscript:

-The title needs to be changed, due to the low number of samples, in my opinion, this is just a pilot study.

-The main influencing factor on the isotope composition of the milk/casein is the cow feed. However, this is not explained  in the introduction in any way. Later on it is mentioned in the discussion (and that general part concerning the feed needs to be moved to the introduction). Information concerning the agricultural practice of the sampled farms needs to be moved to the "Materials and Methods" section (pasture fed).

-As one sample is regarded as outlier and explained by the dominant use of maize feed, what control/Information do the authors actually have about what has been fed to the cows?  

-Based on the available 15N data there seems to me an over-interpretation with respect to correlation of the results to climate, when taking into account the many other possibilities for explanation (see line 268).

-Another very important factor that is not sufficiently discussed and introduced is the seasonality. This is also not discussed at all, merely mentioned as explanation for results not in Agreement with literature data..

-The data for all samples (just 16) is not presented. They need to be given in the manuscript in a table, together with locality, date of sampling and further information available (e.g. organic).

-Figures showing the samples as in Fig. 3 for each investigated element should be provided, to enable the reader to see potential correlation of isotope ratios and geographic origin. As topography is regarded by the authors as as important influencing factor, it might be instructive to Show the respective isotope data with a topographic map.   

As mentioned above, I regard the presented study as a pilot-study to evaluate feasibility of the isotope method for control of geographic origin. In that way, the study might be justified and the manuscript worth to be published, but the results, respective interpretation and discussion needs to be modified in that context.

Author Response

Reviewer 1

Comments and Suggestions for Authors

The authors investigate 16 Casein samples extracted from milk from 16 different dairy farms in New Zealand during late summer early autumn. They investigate the stable isotope ratios and interpret them with respect to geographic origin.

I see several shortcomings in the manuscript:

-The title needs to be changed, due to the low number of samples, in my opinion, this is just a pilot study.

We have changed the topic in toPotential of Casein to Record Stable Isotopic Variation of Cow Milk in New Zealand”

-The main influencing factor on the isotope composition of the milk/casein is the cow feed. However, this is not explained in the introduction in any way. Later on, it is mentioned in the discussion (and that general part concerning the feed needs to be moved to the introduction).

That section has been moved in to Introduction which is now lies in the lines of 49-51 and also mentioned about the feed in the line 43.

Information concerning the agricultural practice of the sampled farms needs to be moved to the "Materials and Methods" section (pasture fed).

Section which previously addressed the general feeding practice of New Zealand in the beginning of Discussion section, has been moved to lines 86-94 in 2.1 Material section.

-As one sample is regarded as outlier and explained by the dominant use of maize feed, what control/Information do the authors actually have about what has been fed to the cows? 

We actually don’t have a control value but since we know that farm provided maize as the majority portion of the feed, our hypothesis is built upon evidence from previous studies. Hence, in the revised version, we have edited our statement in a more hypothetical manner (lines 333-335).

-Based on the available 15N data there seems to me an over-interpretation with respect to correlation of the results to climate, when taking into account the many other possibilities for explanation (see line 268).

Since the agricultural practices were consistent throughout the sampling farms and based on the evidence from a recent study (stated in line 408-409) states milk d15N holds a metabolic origin rather than a digestive origin; we predicted that d15N could be affected by climate parameters as South Island (Sub-Antarctic) and North Island (Temperate) have different climates.

Revised lines (378-380)

-Another very important factor that is not sufficiently discussed and introduced is the seasonality. This is also not discussed at all, merely mentioned as explanation for results not in Agreement with literature data.

This has been introduced in to line 44 and also has been explained in the last paragraph of the discussion section under “Moreover, even though seasonal changes could affect the type of pasture growth (C3 and C4) which then could be observed through the variations in milk/casein d13C, d15N and d18O [3], its effect would be considered insignificant in this study as the sampling process was carried out within a period of less than two weeks

-The data for all samples (just 16) is not presented. They need to be given in the manuscript in a table, together with locality, date of sampling and further information available (e.g. organic).

Due to the commercial sensitivity of the isotopic data, it was not stated in the manuscript. However, date of sampling and locality with the latitudes and longitudes are given using Table 1 (lines 83-84 and lines 108-109)

-Figures showing the samples as in Fig. 3 for each investigated element should be provided, to enable the reader to see potential correlation of isotope ratios and geographic origin. As topography is regarded by the authors as an important influencing factor, it might be instructive to Show the respective isotope data with a topographic map.   

A table of loadings have been added as Table 2. (line 272) and since this was a pilot study we have reconstructed the sentences where topography was mentioned by converting them to give a geographic view.

As mentioned above, I regard the presented study as a pilot-study to evaluate feasibility of the isotope method for control of geographic origin. In that way, the study might be justified and the manuscript worth to be published, but the results, respective interpretation and discussion needs to be modified in that context.

The sentences have been modified and have used the option “Track changes”.

Reviewer 2 Report

I have attached a file.

Please let me know if this didn't appear.

Author Response

Reviewer 2

The authors investigate the use of stable isotopes in the determination of provenance of dairy products in New Zealand. I believe the experiment is well designed, both in terms of data collection and the statistical techniques used. However, I found the results section hard to follow; more information is needed and it needs to be presented a bit more clearly.

In particular:
1. I found the description of the PCs on lines 168 to 182 hard to follow. I believe a table would have helped in quickly seeing the loadings.

A table of loadings have been added as Table 2. (line 272)

  1. The information on lines 183-186 and what Figure 3 shows needs more explanation.

In the revised edition, previous Figure 2 and Figure 3 has become Figure 3 and Figure 4 respectively and we have explained the PCA plots more clearly (lines 231-335).

  1. The authors conclude that the stable isotopes of Oxygen and Hydrogen are related to topography, but this was not illustrated with the observations, i.e. the authors did not show any data analysis based on topography.

Apologies for the confusion, we have corrected the topography term as “Geographical”. Also, we have included the sampling locations (locality), date of sample collection to support the sampling information which is stated in lines 82-85.

Other comments:
1. I believe the data was scaled when PCA was done. If so this should be mentioned in the Methods section, e.g. after Line 132.

It has been revised in line 165 asOur data was scaled to z-scores prior to the Principal Component Analysis

Line 60: “New Zealand provide”, might be better as “New Zealand provides”

Corrected

Line 141: add per mil, after -16.9.

Line 174 – per mil symbol was added

Figure 1: It might be better if the per mil symbol was used in the y-axis label instead of “d”.

corrected

The following figure has been added to demonstrate the geographical patterns in the measured parameters.

Figure 2. Distribution of stable isotopic values measured in casein samples from dairy farms across the North Island (∆) and South Island (○) of New Zealand: (a) d2H values of casein; (b) d18O values of casein; (c) d13C values of casein; (d) d15N values of casein.

Round 2

Reviewer 1 Report

Title: I recommend another title that contains "pilot study" or "feasibility study".

Interpretation of 15N signal: Theoretically, the differences in 15N of North and South Island also might be due to different soil 15N values of the two islands, isn't it? Explain, why you prefer the explanation of the differences in 15N between the two Islands by temperature than soil type.

Seasonality: Already in the Introduction there should be the sentence that seasonality (as mentioned in line 44) is not relevant in the present study, as only fresh milk is investigated and all samples have been collected within a short time span.

Data: I think the data needs to be presented to enable the reader to look at the elemental isotope patterns. As the data only contain 16 samples from a very short time interval and without any proof that the values will be the same the following year (during the same period), commercial sensitivity cannot be reasonably argued against their publication. Instead, if that is feared, the authors should rather remove the geographic coordinates, which are of no use without the combination with the isotope data anyway (and in this way the data cannot be used without the geographic position of the samples and is thus protected). Instead give at least the altitude. Captions of Table 3 are in this regard  incorrect, as the isotope values are not given.

The figures 2A-D should be coloured, the shades of gray are unreadable. Preferentially, also Figs. 3 and 4 should be coloured.

Author Response

Title: I recommend another title that contains "pilot study" or "feasibility study".

Title has been changed to

Feasibility of Casein to Record Stable Isotopic Variation of Cow Milk in New Zealand   

Interpretation of 15N signal: Theoretically, the differences in 15N of North and South Island also might be due to different soil 15N values of the two islands, isn't it? Explain, why you prefer the explanation of the differences in 15N between the two Islands by temperature than soil type.

Soil data added.  Plot in supplementary material and following sentences added to text.

Soil d15N variation could be a major contributor to the observed casein d15N values.  However, our measurements of soil d15N, presented as supplementary material, show a very weak relationship with latitude (r2=-0.05) or with casein d15N (r2=0.12). Hence the soil d15N is a minor driver of casein d15N in this case.

Seasonality: Already in the Introduction there should be the sentence that seasonality (as mentioned in line 44) is not relevant in the present study, as only fresh milk is investigated and all samples have been collected within a short time span.

Sentence added

The sampling represents a snapshot in time and no attempt is made to account for seasonal effects. 

Data: I think the data needs to be presented to enable the reader to look at the elemental isotope patterns. As the data only contain 16 samples from a very short time interval and without any proof that the values will be the same the following year (during the same period), commercial sensitivity cannot be reasonably argued against their publication. Instead, if that is feared, the authors should rather remove the geographic coordinates, which are of no use without the combination with the isotope data anyway (and in this way the data cannot be used without the geographic position of the samples and is thus protected). Instead give at least the altitude. Captions of Table 3 are in this regard  incorrect, as the isotope values are not given

The table has been extended to include all casein isotope values.

The figures 2A-D should be coloured, the shades of gray are unreadable. Preferentially, also Figs. 3 and 4 should be coloured.

Figures are now in colour
